# RN-D: Discretized Categorical Actors for On-Policy Reinforcement Learning

Yuexin Bian[* 1]   Jie Feng[* 1]   Tao Wang[1]   Yijiang Li[1]   Sicun Gao[1]   Yuanyuan Shi[1]

## Abstract

On-policy Reinforcement Learning (RL) remains a dominant paradigm for continuous control, yet standard implementations rely on Gaussian actors and relatively shallow MLP policies, often leading to brittle optimization when gradients are noisy, and policy updates must be conservative. In this paper, we revisit actor policy representation as a first-class design choice for on-policy RL. We study discretized categorical actors, which represent each action dimension as a distribution over discrete bins and induce a policy objective analogous to classification cross-entropy loss. Building on architectural advances from supervised learning, we further pair discretized categorical actors with regularized networks, yielding RN-D. Across diverse continuous-control benchmarks, we show that simply replacing the standard Gaussian actor with our proposed actor substantially improves performance, achieving state-of-the-art results within on-policy RL. We release our code at https://github.com/alwaysbyx/RND-RL.

## 1. Introduction

On-policy reinforcement learning (RL) is a central paradigm for sequential decision-making, underpinning widely used algorithms such as policy gradient methods and their modern variants (Sutton et al., 1998; Schulman et al., 2015a; Mnih et al., 2016b; Schulman et al., 2017). Despite their favorable theoretical properties and widespread adoption, these methods remain notoriously difficult to optimize in practice. For example, Proximal Policy Optimization (PPO) is often reported to exhibit unstable training dynamics, pronounced sensitivity to hyperparameters, and strong dependence on policy parameterization choices (Andrychowicz

et al., 2020). Even on standard continuous-control benchmarks, achieving reliable performance typically requires careful "code-level" optimization (Huang et al., 2022a). These observations suggest that the primary challenges of on-policy RL are not solely due to insufficient data or model capacity, but are closely tied to the optimization behavior induced by architectural and parameterization choices.

Tremendous efforts have been devoted to understanding and improving the training performance of on-policy reinforcement learning algorithms, including the development of improved advantage estimators (Schulman et al., 2015c), analyses highlighting the role of value estimation (Wang et al., 2025), and the design of constrained policy update mechanisms (Schulman et al., 2015a; 2017; Xie et al., 2024). In contrast, the parameterization of the policy itself is often treated as a secondary design choice, with continuous control tasks typically adopting Gaussian policies with diagonal covariance as a default since (Williams, 1992), largely due to its analytical and computational convenience.

Recent work has begun to challenge the conventional from two complementary directions. First, earlier on-policy studies show that discretizing continuous actions and optimizing categorical policies can yield substantial gains over Gaussian actors (Tang & Agrawal, 2020; Zhu et al., 2024). Second, value-function learning has increasingly embraced classification-style objectives: distributional RL trains categorical return distributions with a cross-entropy loss (Bellemare et al., 2017), and recent work shows that replacing MSE with cross-entropy can improve robustness to noisy bootstrapped targets and support scaling to higher-capacity networks (Farebrother et al., 2024; Hafner et al., 2025). These advances are particularly pronounced in off-policy RL, where the value function is the primary object being optimized and directly shapes the policy update. A symmetric perspective applies to on-policy RL: the actor is the object being optimized, and discretized categorical actors turn the policy update into a cross-entropy-like objective over action bins, making architectural and optimization choices that benefit classification especially relevant. A closely related phenomenon appears in on-policy RL for language models, where the policy is categorical over tokens and PPO-style optimization is widely used for RLHF (Yue et al., 2025; Guo et al., 2025; Yu et al., 2025; Sheng et al., 2025). Moreover, on the theoretical side, (Agarwal et al., 2021) shows

[*]Equal contribution  [1]University of California San Diego, La Jolla, USA. Correspondence to: Yuexin Bian <yubian@ucsd.edu>.

*Proceedings of the 43rd International Conference on Machine Learning*, Seoul, South Korea. PMLR 306, 2026. Copyright 2026 by the author(s).

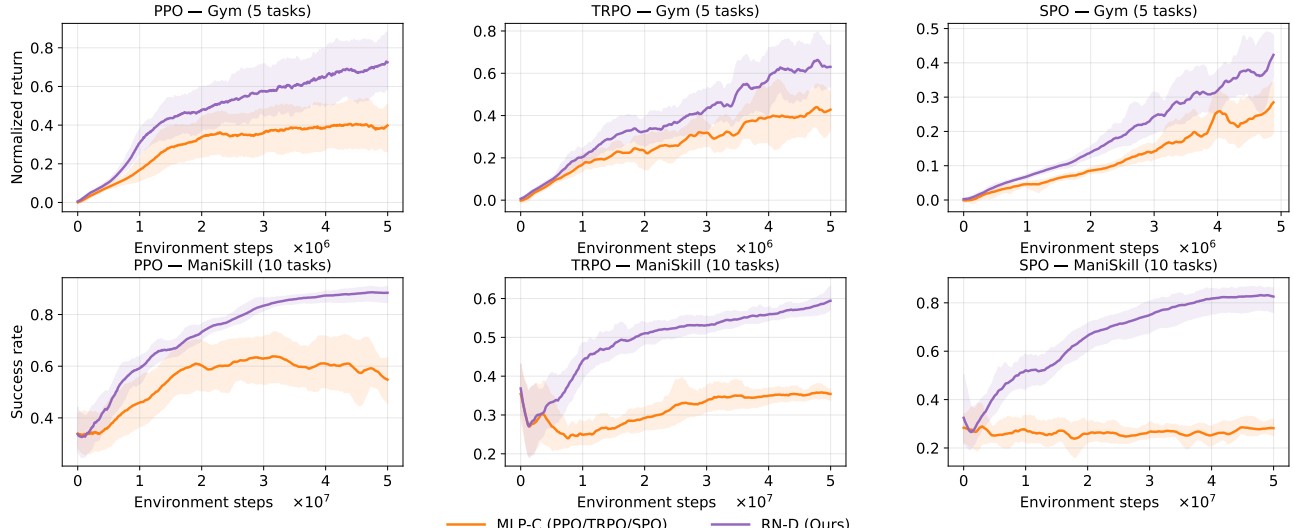

*Figure 1.* **RN-D consistently improve on-policy RL across algorithms.** We compare the dominant standard implementation, a continuous MLP actor (MLP-C), with our regularized discretized actor (RN-D) under PPO, TRPO, and SPO on Gym locomotion and ManiSkill manipulation benchmarks. RN-D achieves faster convergence and higher final performance across both algorithms and benchmarks.

that policy gradient methods admit global convergence guarantees with the softmax policy parameterization under the condition of exact gradients and adequate exploration.

*Motivated by this connection, we revisit actor design in on-policy RL for continuous control tasks through the lens of classification-style objectives.* We make a simple drop-in change to the policy parameterization: Instead of a Gaussian policy with continuous action outputs, we use a factorized categorical policy, which turns the on-policy policy loss into a cross-entropy-like objective over selected bins. This formulation enables the actor update to inherit optimization properties commonly associated with classification, while representing continuous control through discretized action dimensions. We then pair this parameterization with regularized actor networks, which promote more stable training.

Empirically, this simple change leads to substantial and consistent improvements over the dominant standard implementation, a continuous MLP Gaussian actor. As shown in Figure 1, our regularized discretized actor achieves faster convergence and higher final performance on both Gym locomotion and ManiSkill manipulation benchmarks. The same trend holds across PPO (Schulman et al., 2017), TRPO (Schulman et al., 2015b), and SPO (Xie et al., 2024), suggesting that the benefit is not tied to a particular on-policy surrogate objective, but instead reflects a more general advantage of combining classification-style actor parameterization with regularized network design.

The contributions of this paper can be summarized as:

- We revisit actor parameterization and architecture as

underexplored levers for improving on-policy continuous control, and study discretized categorical policies whose optimization resembles cross-entropy loss.

- We propose a regularized actor network design for discretized on-policy learning that improves optimization stability and reduces gradient variance during training.

- We demonstrate that our approach significantly improves RL performance and accelerates convergence across standard locomotion and ManiSkill benchmarks, covering both state-based and vision-based tasks. Moreover, the method is algorithm-agnostic and consistently benefits multiple on-policy algorithms.

## 2. Related Works

**On-policy reinforcement learning.** On-policy policy gradient methods optimize the expected return by differentiating through a stochastic policy, dating back to REINFORCE (Williams, 1992) and the policy gradient theorem with function approximation (Sutton et al., 1999). Modern deep actor–critic systems (e.g., A3C) improved stability and throughput via parallel data collection and actor–critic updates (Mnih et al., 2016a). A central challenge in on-policy learning is balancing stability and sample reuse. Natural-gradient and trust-region perspectives (Cobbe et al., 2021; Wu et al., 2017) motivated algorithms that constrain policy updates, most notably TRPO (Schulman et al., 2015b). PPO (Schulman et al., 2017) popularized a simpler clipped surrogate objective that supports multiple epochs of minibatch updates, and generalized advantage estimation (GAE) further improved variance–bias tradeoffs in advantage com-

putation (Schulman et al., 2015c).

In parallel, several large-scale empirical studies have shown that the performance of on-policy methods depends strongly on "implementation ingredients" beyond the high-level objective, including normalization, clipping, network design, and optimization details (Andrychowicz et al., 2020; Gronauer et al., 2021). Motivated by these findings, our work targets a complementary axis: we study how the *policy representation itself*, in particular categorical parameterizations with neural network architecture design, affects optimization behavior under standard on-policy training.

**Actor policy parameterization for continuous control.** Most continuous-control actor–critic implementations parameterize the policy with a diagonal Gaussian distribution (Andrychowicz et al., 2020) due to its simplicity and reparameterization-friendly sampling; however, this choice can interact with exploration and gradient variance. To address the limitations of Gaussian policies with bounded action spaces, prior work replaces them with the finite-support Beta distribution (Chou et al., 2017), yielding a bias-free, lower-variance policy that performs well empirically. Further, considering the extremes along each action dimension, a Bernoulli distribution is applied to replace the Gaussian parametrization of continuous control methods and show improved performance on several continuous control benchmarks (Seyde et al., 2021). Within PPO-style training, PPO-CMA adapts the exploration covariance inspired by CMA-ES to mitigate premature variance collapse in Gaussian policies (Hämäläinen et al., 2020).

An orthogonal line of work replaces continuous distributions with *discretized* categorical policies over per-dimension action bins. Tang and Agrawal (Tang & Agrawal, 2020) showed that factorized categorical policies can scale to high-dimensional control and often improve on-policy optimization, and further proposed ordinal parameterizations to explicitly encode the natural ordering of action bins. More recently, unimodal discrete parameterizations (e.g., Poisson-based) were introduced to enforce ordering structure and to reduce undesirable multimodality and gradient variance in discretized actors (Zhu et al., 2024).

**Network architectures.** Recent advances in computer vision and NLP have been largely driven by scaling model capacity (Lee et al., 2024). In supervised learning, architectural choices, including skip connections, normalization, and depth/width scaling, are known to strongly influence optimization. Residual networks (He et al., 2016) improve the trainability of deep models via identity pathways, while batch normalization (Ioffe & Szegedy, 2015) and layer normalization (Ba et al., 2016) help stabilize activations and gradients. In contrast, network design and systematic scaling have been comparatively less explored in deep rein-

forcement learning, where many standard benchmarks and implementations (Huang et al., 2022b; Andrychowicz et al., 2020) still rely on relatively shallow MLP backbones (e.g., 2-3 layers) for the actor and critic.

Several recent works revisit careful network designs for deep RL, often highlighting how network capacity can substantially influence performance (Lee et al., 2024; Hansen et al., 2024; 2025). SimBa (Lee et al., 2024) and its follow-up SimBa-v2 (Lee et al., 2025) propose scalable MLP backbones for deep RL, showing that residual pathways and improved normalization can unlock consistent performance gains as model capacity increases. BroNet provides an extensive study of critic scaling and introduces a regularized residual MLP architecture to unlock performance gains when increasing critic capacity (Nauman et al., 2025). While these works primarily improve performance by scaling backbones and strengthening value learning (often in off-policy settings), a systematic understanding of how *actor* network architecture influences *on-policy* optimization remains less mature. We complement this line of research by studying the actor network design principles under on-policy training, and by considering discretized categorical actors whose policy objective resembles a cross-entropy loss over action bins.

## 3. Preliminaries

We consider standard episodic reinforcement learning in continuous-control Markov decision processes (MDPs) with state space $\mathcal{S}$, action space $\mathcal{A} \subset \mathbb{R}^m$, transition dynamics $P(\cdot \mid s, a)$, reward function $r(s, a)$, and discount factor $\gamma \in (0, 1)$. A (stochastic) actor policy $\pi_\theta(a \mid s)$ is parameterized by $\theta$ and induces the discounted return

$$J(\theta) \triangleq \mathbb{E}_{\tau \sim \pi_\theta} \Big[ \sum_{t=0}^{\infty} \gamma^t r(s_t, a_t) \Big], \tag{1}$$

where $\tau = (s_0, a_0, s_1, a_1, \dots)$ is a trajectory generated by interacting with the environment under $\pi_\theta$.

### 3.1. On-policy actor optimization

On-policy methods such as Proximal Policy Optimization (PPO) (Schulman et al., 2017) update the actor using data collected from the current policy. Let $\pi_{\theta_{\text{old}}}$ denote the behavior policy used to sample a batch of trajectories. PPO maximizes a clipped surrogate objective that stabilizes policy updates by constraining the change in action probabilities:

$$\mathcal{L}^{\text{PPO}}(\theta) = \mathbb{E}\Big[ \min \big(r_t(\theta)\hat{A}_t, \ \text{clip}(r_t(\theta), 1-\epsilon, 1+\epsilon)\hat{A}_t\big) \Big]. \tag{2}$$

where $r_t(\theta) \triangleq \frac{\pi_\theta(a_t|s_t)}{\pi_{\theta_{\text{old}}}(a_t|s_t)}$ is the importance ratio, $\epsilon > 0$ is the clipping threshold, and $\hat{A}_t$ is an advantage estimate computed from a learned value function $V_\phi(s)$ (e.g., using

generalized advantage estimation). Unless stated otherwise, we consider the standard actor–critic setup where $\phi$ is updated by minimizing a squared-error value loss, and the actor is updated by maximizing (2).

### 3.2. Policy parameterizations for continuous control

In this work, we consider two common stochastic actor families for continuous control: a continuous Gaussian policy and a discrete categorical policy constructed via action discretization. Without loss of generality, we assume the action space $\mathcal{A} = [-1, 1]^m$.

**Continuous Gaussian actor.** The continuous actor models $\pi_\theta(a \mid s)$ as a multivariate Gaussian distribution

$$\pi^c(a \mid s) = \mathcal{N}\big(\mu_\theta(s), \Sigma_\theta(s)\big), \tag{3}$$

where $a \in \mathbb{R}^m$ is an $m$-dimensional continuous action vector. The mean $\mu_\theta(s) \in \mathbb{R}^m$ is produced by a neural network. The covariance is typically chosen to be diagonal,

$$\Sigma_\theta(s) = \mathrm{Diag}\big(\sigma_\theta^2(s)\big) \in \mathbb{R}^{m \times m}, \tag{4}$$

with $\sigma_\theta(s) \in \mathbb{R}_+^m$ parameterized by $\theta$ (either state-dependent or state-independent).

**Discrete categorical actor via uniform discretization.** In contrast to continuous parameterizations, a discrete actor represents the policy as a categorical distribution over a finite set of discretized actions. Without loss of generality, we assume $\mathcal{A} = [-1, 1]^m$. For each action dimension $i \in \{1, \ldots, m\}$, we define a uniform discretization of $[-1, 1]$ into $K$ bins:

$$\mathcal{A}_i = \left\{ \frac{2j}{K-1} - 1 \;\middle|\; j = 0, 1, \ldots, K-1 \right\}. \tag{5}$$

The policy over these $K$ discrete choices is parameterized by logits $z_\theta(s)_{i,j}$ and a softmax:

$$\pi^d(a_j^i \mid s) = \frac{\exp\big(z_\theta(s)_{i,j}\big)}{\sum_{k=1}^K \exp\big(z_\theta(s)_{i,k}\big)}, \quad j = 1, \ldots, K. \tag{6}$$

We assume conditional independence across action dimensions, yielding a factorized joint policy

$$\pi^d(a \mid s) = \prod_{i=1}^m \pi^d(a_{j_i}^i \mid s), \tag{7}$$

where $a = (a_{j_1}^1, \ldots, a_{j_m}^m)$ corresponds to selecting one bin index $j_i$ per dimension. The resulting action lies in the original range $[-1, 1]^m$ by construction.

Note that beyond the plain softmax over per-bin logits, some works (Tang & Agrawal, 2020; Zhu et al., 2024) explore structured discrete actor parameterizations that explicitly

exploit the ordinal structure of discretized action bins to improve optimization stability. In this paper, we instead focus on the simplest baseline: uniform discretization with an independent softmax categorical policy per action dimension, so as to isolate the effect of discretizing the actor without additional architectural constraints.

**Training with PPO.** Both the continuous Gaussian actor and the discretized categorical actor can be trained under the same on-policy optimization framework (e.g., PPO). The only policy-specific ingredient is the log-likelihood $\log \pi_\theta(a_t \mid s_t)$ used to form the importance ratio in the PPO surrogate objective: it is evaluated under a Gaussian density for $\pi^c$ and under a categorical probability mass function for $\pi^d$. All other components of the training pipeline, on-policy rollout collection, advantage estimation, and clipped surrogate maximization can be identical.

## 4. Revisiting Discrete Categorical Policies

This section revisits discretized categorical policies for continuous control from two complementary viewpoints. First, we analyze how the policy-gradient estimator behaves under a factorized categorical policy and how its variance depends on the discretization. Second, we provide an optimization perspective that connects PPO actor updates to standard supervised losses: cross-entropy for categorical actors and squared-error (under Gaussian likelihood) for continuous actors. These perspectives motivate the empirical and algorithmic choices studied in later sections.

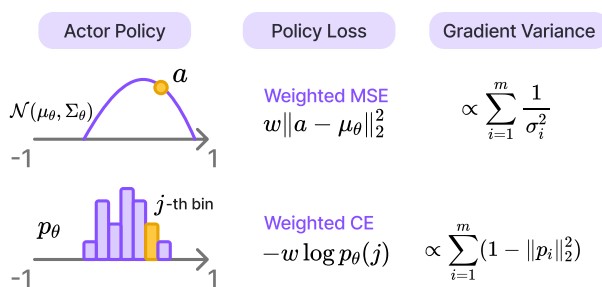

*Figure 2.* Understanding Continuous Gaussian and Discrete categorical actor policy from two perspectives.

### 4.1. Policy-gradient estimator and variance

**On-policy policy gradient.** To analyze the stochastic gradient structure underlying PPO, we consider the policy-gradient term obtained by differentiating the loss terms in (2) with respect to $\theta$:

$$g(\theta) \triangleq \mathbb{E}_{(s_t, a_t) \sim \pi_{\theta_{\mathrm{old}}}} \Big[ w_t(\theta) \, \hat{A}_t \, \nabla_\theta \log \pi_\theta(a_t \mid s_t) \Big], \tag{8}$$

where $w_t(\theta)$ denotes the effective PPO importance weight induced by clipping,

$$w_t(\theta) = \begin{cases} r_t(\theta), & \text{if unclipped}, \\ 0, & \text{if clipped}, \end{cases} \tag{9}$$

$r_t(\theta)$ is the importance ratio and $\hat{A}_t$ is an advantage estimator. In practice, (8) is estimated from finite on-policy rollouts, so its optimization behavior is strongly influenced by the variance of the random vector $w_t(\theta)\,\hat{A}_t\,\nabla_\theta \log \pi_\theta(a_t \mid s_t)$.

**A vanilla policy-gradient toy setting.** To isolate the intrinsic stochasticity of the score function, we consider a simplified one-step on-policy setting with a fixed state $s$ and the vanilla REINFORCE estimator (i.e., $w_t(\theta) \equiv 1$). We further assume a constant return $R_t \equiv R$ independent of the sampled action (Zhu et al., 2024). The stochastic gradient estimator becomes

$$\hat{g}(\theta) = R\,\nabla_\theta \log \pi_\theta(a \mid s), \quad a \sim \pi_\theta(\cdot \mid s). \tag{10}$$

Although the true gradient is zero in this setting (since $R$ does not depend on $a$), the estimator (10) generally has nonzero variance; this is precisely the variance that baselines/advantages are designed to reduce in practice.

**Proposition 4.1** (Gradient variance for Gaussian vs. categorical policies). *Fix a state $s$ and consider the one-step REINFORCE estimator $\hat{g}(\theta) = R\,\nabla_\theta \log \pi_\theta(a \mid s)$ with $a \sim \pi_\theta(\cdot \mid s)$ and constant return $R$. Then the conditional covariance for the two policy families are:*

*[1] **Gaussian (gradient w.r.t. mean).** Denote $\pi^c(a \mid s) = \mathcal{N}(\mu, \Sigma)$ with diagonal $\Sigma = \mathrm{Diag}(\sigma^2)$ and treat $\mu \in \mathbb{R}^m$ as the parameter (with $\Sigma$ fixed). Then*

$$\mathbb{E}\big[\|\hat{g}_\mu\|_2^2 \mid s\big] = R^2\,\mathrm{Tr}(\Sigma^{-1}) = \sum_{i=1}^m \frac{R^2}{\sigma_i^2}. \tag{11}$$

*[2] **Categorical (gradient w.r.t. logits).** Denote the discretized categorical policy factorize across $m$ action dimensions as $\pi^d(a \mid s) = \prod_{i=1}^m \pi^d(a_{j_i}^i \mid s)$, where $p_i(s) \in \Delta^{K-1}$ is the per-dimension softmax probability vector over $K$ bins and $j_i \sim \mathrm{Cat}(p_i(s))$. Treat the per-dimension logits $z_i(s) \in \mathbb{R}^K$ as parameters and let $z(s) = [z_1(s); \ldots; z_m(s)] \in \mathbb{R}^{mK}$. Then*

$$\mathbb{E}\big[\|\hat{g}_z\|_2^2 \mid s\big] = R^2 \sum_{i=1}^m \Big(1 - \|p_i(s)\|_2^2\Big) \leq mR^2\Big(1 - \frac{1}{K}\Big) \tag{12}$$

*Moreover, the inequality is tight if and only if $p_i(s)$ is uniform over the $K$ bins for all $i$.*

*Proof.* The full proof is provided in Appendix A. □

**Insight:** For continuous Gaussian actors, gradient variance can grow sharply as the policy standard deviation decreases, a behavior that commonly occurs in practice as exploration is reduced over the course of training. Even at early training stages, where the standard deviation remains relatively large, e.g., $\sigma = 1$, the per-dimension gradient-variance gap between Gaussian and categorical policies is lower bounded by $1/K$, and thus the total gap scales linearly with the action dimension $m$.

## 4.2. Optimization view: cross-entropy vs. squared error

**Gaussian actor: (weighted) squared error.** For a diagonal Gaussian with (state-dependent or state-independent) variance, the negative log-likelihood is

$$-\log \pi_\theta^c(a \mid s) = \frac{1}{2}(a - \mu_\theta(s))^\top \Sigma_\theta(s)^{-1}(a - \mu_\theta(s)) + \frac{1}{2}\log|\Sigma_\theta(s)| + \text{constant}. \tag{13}$$

If $\Sigma_\theta(s) = \sigma^2 I$ is fixed, then minimizing (13) with respect to $\mu_\theta(s)$ is equivalent to minimizing a mean squared error:

$$-\log \pi_\theta^c(a \mid s) \propto \|a - \mu_\theta(s)\|_2^2. \tag{14}$$

In combination with equation (8), the Gaussian actor update admits an interpretation as a weighted mean squared error regression, where the weights are determined by the advantage and the importance ratio.

**Categorical actor: (weighted) cross-entropy.** For a discretized categorical policy, a sampled action corresponds to a bin index $j_i$ per dimension. The negative log-likelihood (NLL) for each dimension is

$$-\log \pi_\theta^d(a^i \mid s) = -\log p_i(j_i \mid s), \tag{15}$$

which is exactly the cross-entropy between a one-hot target $e_{j_i}$ and the predicted distribution $p_i(s)$. Under equation (8), the objective reduces to a weighted cross-entropy loss, where the weights are given by the advantage-modulated importance ratio (with clipping). Accordingly, the categorical actor update corresponds to minimizing this weighted cross-entropy over the sampled action bins.

With strong empirical results to support the use of cross-entropy as a "drop-in" replacement for the mean squared error (MSE) regression loss in deep RL for value function learning (Farebrother et al., 2024), few works address this for the actor network.

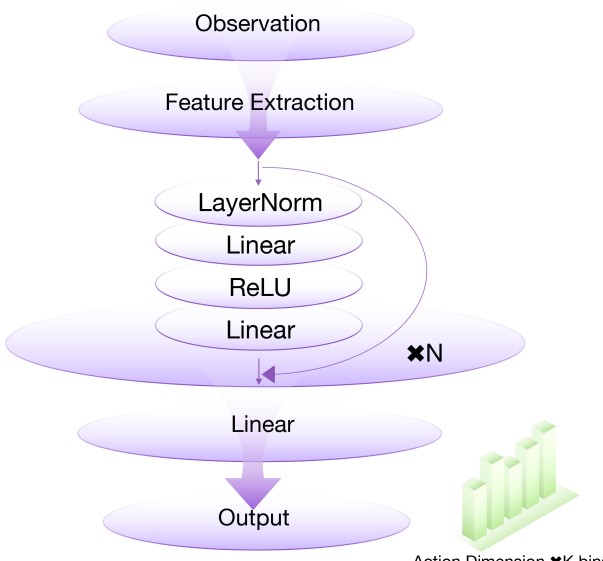

*Figure 3.* Proposed **R**egularized **N**etwork for **D**iscrete action policies (RN-D). The actor consists of a feature extractor (MLP or CNN) followed by pre-LayerNorm residual MLP blocks, enabling stable optimization and improved scalability. The output layer produces categorical logits over $K$ bins for each action dimension.

---

**Insight:** From this perspective, On-policy RL training differs across policy families primarily through the likelihood model that defines $\log \pi_\theta(a \mid s)$: categorical policies induce (weighted) cross-entropy objectives over bins, while Gaussian policies induce (weighted) quadratic objectives over continuous action.

---

Although prior work provides strong empirical evidence that cross-entropy can replace the mean squared error (MSE) regression loss for value function learning in deep reinforcement learning (Farebrother et al., 2024; Lee et al., 2025), the consequences of analogous loss choices for actor network learning remain largely unexplored. Motivated by viewing the policy objective through this loss-based lens: weighted MSE for Gaussian actors versus weighted cross-entropy for categorical actors, we hypothesize that the **classification**-style structure of the categorical update can better exploit neural network capacity. This perspective further motivates our use of discrete categorical actors as a scalable design choice for training large RL policies.

## 5. Actor Network Architecture

We adopt a unified actor network architecture for discrete categorical policies that emphasizes optimization stability and scalability across high-dimensional action spaces. The architecture is composed of a feature extraction stage, a stack of residual feedforward blocks, and a final output projection to categorical action logits.

**Feature Extraction.** The actor first maps raw observations to a latent feature representation using a task-dependent encoder. For low-dimensional state inputs, we employ a multi-layer perceptron (MLP) encoder composed of linear layers and nonlinear activations. For high-dimensional observations (e.g., images), a convolutional neural network (CNN) encoder is used instead. The encoder produces a shared latent representation that is consumed by subsequent feedforward blocks.

**Residual Feedforward Blocks.** Following feature extraction, the actor applies a stack of $N$ residual feedforward blocks. Each block is equipped with a residual connection,

$$\mathbf{h}_{\ell+1} = \mathbf{h}_\ell + \mathrm{FFN}(\mathrm{LN}(\mathbf{h}_\ell)),$$

where $\mathrm{LN}(\cdot)$ denotes layer normalization. We employ pre-layer normalization by applying LayerNorm before each feedforward block, a design that improves optimization stability in deep architectures, including Transformer models (Xiong et al., 2020). Following (Vaswani et al., 2017), the feedforward network adopts an inverted bottleneck structure. The hidden dimension is first expanded from $d_h$ to $4d_h$ and passed through a ReLU activation, and a second linear layer projects the representation back to $d_h$.

**Output Layer.** The final hidden representation is passed through a linear projection that outputs logits for a discrete categorical distribution. Specifically, for an action space with $d$ dimensions and $K$ bins per dimension, the output layer produces $d \times K$ logits, which parameterize independent categorical distributions over each action dimension.

**Relation to Prior Architectures.** The proposed actor architecture shares structural similarities with several existing designs. Residual MLPs with normalization have been explored in prior reinforcement learning works such as SimBa (Lee et al., 2024; Lee et al.) and BroNet (Nauman et al., 2024), as well as in the feedforward sublayers of Transformer models (Vaswani et al., 2017). Our contribution does not lie in introducing a novel network module, but rather in demonstrating that this architectural choice is particularly well-suited for discrete categorical actor policies. As shown in our experiments, combining our network with categorical action parameterization leads to improved optimization behavior compared to conventional shallow MLP actors commonly used in on-policy RL.

## 6. Experiment

We evaluate the proposed discrete policy with a regularized network (RN-D) under PPO across a diverse set of locomotion and manipulation benchmarks. A full list of tasks, environment details, and hyperparameters is provided in Appendix B.

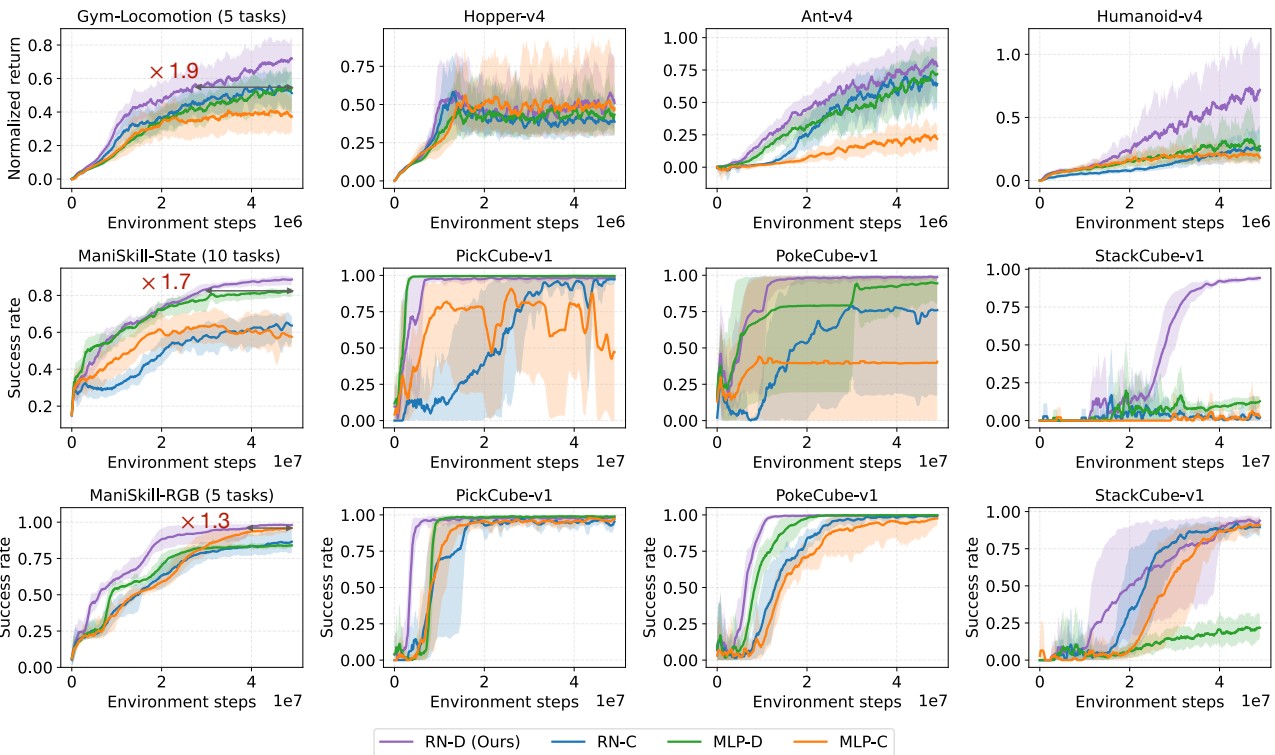

*Figure 4.* Aggregate learning curves across benchmarks. Each subplot reports the mean performance over tasks within a benchmark (MuJoCo locomotion: normalized return; ManiSkill: success rate) as a function of environment steps. Curves are averaged over 5 random seeds; shaded regions denote 95% stratified bootstrap confidence intervals. The red annotations indicate a sample-efficiency speedup.

**Experiment Setup.** We evaluate our methods on three environment families: Gym locomotion (Brockman et al., 2016), ManiSkill (Tao et al., 2025) with state-based observations, and ManiSkill with RGB-based observations. These tasks cover a range of control challenges, including locomotion and dexterous manipulation, varying embodiments, and different observation modalities. Specifically, we consider 5 standard Gym locomotion tasks, 10 ManiSkill manipulation tasks with low-dimensional state observations, and 5 ManiSkill tasks with image-based observations, where policies operate on raw RGB inputs via a convolutional encoder.

**Baselines.** All experiments are conducted with Proximal Policy Optimization (PPO) (Schulman et al., 2017), using implementations adapted from the CleanRL (Huang et al., 2022b) and ManiSkill codebases (Tao et al., 2025). We compare **RN-D**, our regularized-network discrete (categorical) actor, against three baselines: (i) **RN-C**, a continuous policy parameterized by a factorized Gaussian with the same regularized network; (ii) **MLP-C**[1], a continuous factorized-Gaussian policy with a standard MLP actor; and (iii) **MLP-D**, a discrete (categorical) policy with a standard MLP actor. Across all four settings, we keep the critic (value

---

[1]**MLP-C** refers to the widely adopted PPO actor in standard implementations.

network) architecture identical to ensure a fair comparison. For discrete actors, we use $K = 41$ bins throughout.

**Metrics.** To aggregate performance across benchmarks, we use task-standard metrics and normalize returns when needed. For Gym MuJoCo locomotion, we report TD3-normalized return (Fujimoto et al., 2018). For ManiSkill benchmarks, we report success rate, i.e., the fraction of evaluation episodes that achieve the task goal.

**Overall Performance.** Figure 4 shows that **RN-D (ours)** consistently achieves the best performance across both locomotion and ManiSkill. On Gym locomotion, RN-D attains higher normalized returns than the Gaussian baselines, with clear improvements on harder tasks such as Ant and Humanoid. On ManiSkill, RN-D yields the highest average success on the 10 state-based tasks and is the only variant that reliably solves the challenging StackCube task, where other methods remain near-zero success for most of the training. The same trend holds for vision-based ManiSkill tasks: RN-D maintains its advantage when paired with CNN encoders and reaches near-saturated success more reliably than baselines. Beyond final performance, RN-D also reaches strong performance with fewer environment steps. The red annotations indicate a sample-efficiency speedup: RN-D

matches the best baseline's performance using roughly 1.3–1.9× fewer interaction steps. Comparing ablations reveals complementary effects. For state-based settings (locomotion and ManiSkill-State), RN-C versus MLP-C (blue/orange) shows that the regularized backbone improves performance even with Gaussian actors, while MLP-D versus MLP-C (green/orange) indicates that discretization alone can also be beneficial. For vision-based tasks, the CNN encoder provides a strong representation and narrows the gap among actor variants; in some cases MLP-C is already competitive. Nevertheless, RN-D still achieves the best final performance and consistently improves sample efficiency.

**Gradient Variance.** We plot the policy-gradient variance (log scale) on five MuJoCo tasks as a representative illustration (Fig 5). Across training, the Gaussian policy (RN-C/MLP-C) exhibits consistently larger variance, often by orders of magnitude, than its discrete categorical counterpart (RN-D/MLP-D). This disparity is most pronounced for the standard MLP actor: MLP-C (orange) stays far above MLP-D (green) and increases steadily with environment steps, which is consistent with the Gaussian actor's standard deviation shrinking over training and amplifying gradient variability. Replacing the MLP with a regularized network reduces variance for both policy classes. Overall, the discrete categorical policy with a regularized network (RN-D, purple) achieves the lowest policy-gradient variance throughout training.

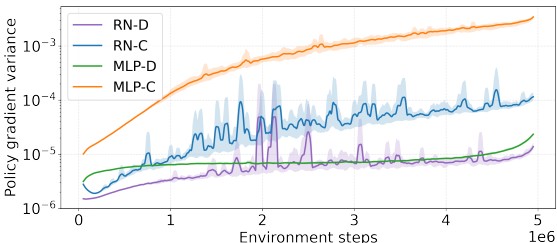

*Figure 5.* The evolution of the policy-gradient variance over training (log scale).

**Component Analysis.** We visualize the gradient signal-to-noise ratio (SNR) and normalized return on Gym locomotion tasks, where SNR is computed as the squared norm of the mean policy gradient divided by its variance across minibatches during training. Following prior work (Andrychowicz et al., 2020), we report the 95th-percentile performance over training, and summarize gradient signal quality using the mean SNR. Figure 6(a) shows that residual connections and layer normalization each improve SNR relative to a plain MLP actor. Figure 6(b) shows a consistent trend in performance, suggesting a strong correlation between gradient signal quality (SNR) and normalized return across architectures.

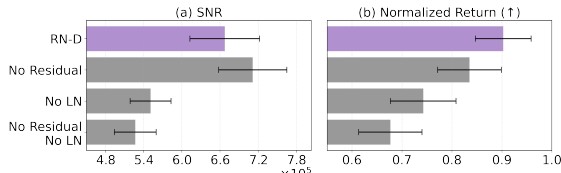

*Figure 6.* Component Analysis. (a) Gradient signal-to-noise ratio (SNR) on Gym locomotion tasks. Bars show the mean and error bars denote one standard deviation across tasks. (b) Average normalized return. Higher SNR correlates with higher returns, with RN-D achieving the best performance.

## 7. Extended Study

**Additional Comparisons with off-policy RL Methods.** In Appendix C.1, we further compare our method to strong off-policy RL baselines, TD3 (Fujimoto et al., 2018) and TD-MPC2 (Hansen et al., 2024), on Humanoid-v4 and StackCube-v1. Our goal is not to claim state-of-the-art performance among all RL algorithms, but to contextualize the practical efficiency of our on-policy actor design against representative strong off-policy methods. As shown in Figure 7(a), the proposed approach achieves strong wall-clock efficiency and performs well on both tasks. In contrast, TD-MPC2 is considerably less computationally efficient in our setting, while TD3 performs well on Humanoid-v4 but struggles on StackCube-v1.

**Role of Cross-Entropy-Like Optimization.** We introduce a loss-swap ablation in Appendix C.2. The proposed discrete actor continues to output logits over $K$ discretized bins per action dimension, but instead of sampling from the categorical distribution, we compute the expected action as the probability-weighted average of bin centers and treat it as the mean of a Gaussian policy. Policy updates are then performed using the standard Gaussian log-likelihood, resulting in a weighted squared-error objective. Figure 7 (b) compares RN-D, RN-C, and the loss-swapped variant. These results show that discretizing the action space alone is insufficient to explain the observed gains. Even with the same discretized representation, switching from a cross-entropy-like objective to a squared-error objective eliminates most of the performance improvement, highlighting the central role of the categorical likelihood objective.

**Scaling the actor network.** In Appendix C.3, we keep the critic fixed and scale the RN-D actor by varying its hidden width from $d_h = 64$ to $512$. We find that wider actors consistently converge faster on both Humanoid-v4 and StackCube-v1, indicating that increased capacity in the proposed RN-D policy can be effectively leveraged to improve learning speed under the same PPO training protocol.

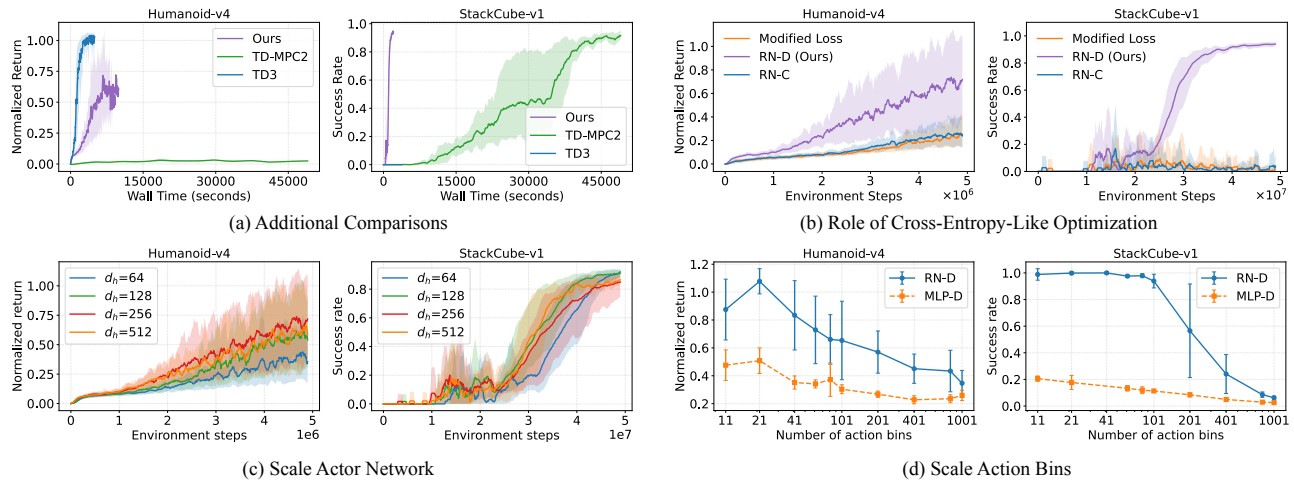

*Figure 7.* Extended studies on sample efficiency, optimization objective, and scaling.

**Scaling the discrete bins.** In Appendix C.4, we conduct an ablation study on discretization granularity by varying the number of action bins from 11 to 1001, with 41 bins used in the main experiments. We find that moderate bin counts generally perform best, while performance often degrades as the number of bins becomes very large. The effective range can also depend on the task: in our experiments, performance is often stable around 11–101 bins, but the optimal choice varies across environments. Overall, discrete-policy performance is sensitive to bin granularity: small-to-moderate bin counts preserve the optimization benefits of discretization, whereas very large bin counts require overly fine-grained predictions that may exceed policy capacity and lead to performance degradation.

**Runtime analysis.** RN-D improves performance without adding meaningful runtime overhead. As shown in Appendix C.5, the cost of increasing the bin count $K$ saturates quickly, suggesting that fine-grained discretization is not the main computational bottleneck once $K$ is moderately large. Table 1 further shows that MLP-D and RN-D achieve comparable or slightly better throughput than their continuous counterparts, with similar wall-clock time. These results indicate that discretization improves policy representation while preserving practical training efficiency, making it a simple and scalable drop-in replacement for standard continuous actors.

## 8. Conclusion

We revisited actor design for on-policy reinforcement learning in continuous control and showed that policy parameterization and architecture provide a strong, underexplored lever for improving PPO-style training. Our approach replaces the standard diagonal-Gaussian actor with a discretized categorical policy over action bins and pairs it with a regularized

*Table 1.* Wall-clock time and throughput, measured by samples per second (SPS), averaged across 5 Gym locomotion tasks with 5 seeds each.

| Method | Wall-clock Time (h) | SPS |
| --- | --- | --- |
| MLP-C | 1.65 | 854.4 |
| MLP-D | 1.56 | 925.3 |
| RN-C | 1.75 | 811.1 |
| RN-D | 1.64 | 873.8 |

actor network, while keeping the critic and the underlying on-policy objective fixed. Across locomotion benchmarks and ManiSkill (state- and vision-based) tasks, this simple actor-side change consistently improves final performance, accelerates convergence, and yields more stable learning.

Several directions appear promising. First, future work can further extend our study beyond PPO by evaluating discretized categorical actors under a broader set of on-policy algorithms. Second, while this work isolates actor-side effects by keeping the critic fixed, understanding actor–critic interactions remains an important direction. Jointly scaling or regularizing both the actor and critic may reveal more symmetric design principles for stable on-policy optimization. Finally, we envision applying this approach to more complex embodied settings, including long-horizon manipulation, contact-rich control, and multi-object tasks, to better characterize its benefits and limitations in challenging real-world regimes.

## Impact Statement

This paper presents work whose goal is to advance the field of reinforcement learning. There are many potential societal consequences of our work, none of which we feel must be

specifically highlighted here.

## Acknowledgements

We thank Haiwen Yu for the contributions to the framework visualization design. This work was supported in part by the U.S. Department of Energy under Grant DE-SC0025495, the Schmidt Sciences AI2050 Early Career Fellowship, and the National Science Foundation under Grant No. 2442689.

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

# A. Proof for Proposition 4.1

*Proof.* Fix a state $s$ and write $\hat{g}(\theta) = R \nabla_\theta \log \pi_\theta(a \mid s)$ with $a \sim \pi_\theta(\cdot \mid s)$ and constant $R$. In both cases below we use the standard identity

$$\mathbb{E}\big[\nabla_\theta \log \pi_\theta(a \mid s) \mid s\big] = \nabla_\theta \int \pi_\theta(a \mid s)\, da = \nabla_\theta 1 = 0, \tag{16}$$

so $\mathbb{E}[\hat{g} \mid s] = 0$ and therefore the conditional covariance satisfies $\mathrm{Cov}(\hat{g} \mid s) = \mathbb{E}[\hat{g}\hat{g}^\top \mid s]$ and, in particular, $\mathbb{E}[\|\hat{g}\|_2^2 \mid s] = \mathrm{Tr}(\mathrm{Cov}(\hat{g} \mid s))$.

**[1] Gaussian policy (gradient w.r.t. mean).** Let $\pi^c(a \mid s) = \mathcal{N}(\mu, \Sigma)$ with fixed diagonal $\Sigma = \mathrm{Diag}(\sigma^2)$ and parameter $\mu \in \mathbb{R}^m$. The log-density is

$$\log \pi^c(a \mid s) = -\tfrac{1}{2}(a - \mu)^\top \Sigma^{-1}(a - \mu) + C,$$

hence

$$\nabla_\mu \log \pi^c(a \mid s) = \Sigma^{-1}(a - \mu). \tag{17}$$

Therefore $\hat{g}_\mu = R\, \Sigma^{-1}(a - \mu)$ and

$$\begin{aligned}
\mathbb{E}[\|\hat{g}_\mu\|_2^2 \mid s] &= R^2\, \mathbb{E}[(a - \mu)^\top \Sigma^{-2}(a - \mu) \mid s] \\
&= R^2\, \mathrm{Tr}\Big(\Sigma^{-2}\, \mathbb{E}[(a - \mu)(a - \mu)^\top \mid s]\Big) \\
&= R^2\, \mathrm{Tr}(\Sigma^{-2}\Sigma) = R^2\, \mathrm{Tr}(\Sigma^{-1}).
\end{aligned}$$

For diagonal $\Sigma = \mathrm{Diag}(\sigma^2)$, $\mathrm{Tr}(\Sigma^{-1}) = \sum_{i=1}^m \sigma_i^{-2}$, yielding (11).

**[2] Categorical policy (gradient w.r.t. logits).** Consider one action dimension $i$ with logits $z_i \in \mathbb{R}^K$ and softmax probabilities $p_i = \mathrm{softmax}(z_i) \in \Delta^{K-1}$. Let $j_i \sim \mathrm{Cat}(p_i)$ and denote the sampled one-hot vector by $e_{j_i} \in \mathbb{R}^K$. For the log-probability $\log \pi^d(a_{j_i}^i \mid s) = \log p_{i,j_i}$, the softmax score w.r.t. logits is

$$\nabla_{z_i} \log p_{i,j_i} = e_{j_i} - p_i. \tag{18}$$

Hence the per-dimension REINFORCE gradient is $\hat{g}_{z_i} = R\,(e_{j_i} - p_i)$, and its conditional second moment is

$$\begin{aligned}
\mathbb{E}[\|\hat{g}_{z_i}\|_2^2 \mid s] &= R^2\, \mathbb{E}[\|e_{j_i} - p_i\|_2^2 \mid s] \\
&= R^2\, \mathbb{E}[\|e_{j_i}\|_2^2 + \|p_i\|_2^2 - 2\, e_{j_i}^\top p_i \mid s] \\
&= R^2\left(1 + \|p_i\|_2^2 - 2\, \mathbb{E}[p_{i,j_i} \mid s]\right) \\
&= R^2\left(1 + \|p_i\|_2^2 - 2\sum_{k=1}^K p_{i,k}^2\right) = R^2\left(1 - \|p_i\|_2^2\right).
\end{aligned}$$

Now assume the $m$ action dimensions factorize: $\pi^d(a \mid s) = \prod_{i=1}^m \pi^d(a_{j_i}^i \mid s)$, so the full log-probability is the sum across $i$ and the full gradient w.r.t. $z = [z_1; \ldots; z_m]$ is the concatenation of per-dimension gradients: $\hat{g}_z = [\hat{g}_{z_1}; \ldots; \hat{g}_{z_m}]$. Therefore $\|\hat{g}_z\|_2^2 = \sum_{i=1}^m \|\hat{g}_{z_i}\|_2^2$, and

$$\mathbb{E}\big[\|\hat{g}_z\|_2^2 \mid s\big] = \sum_{i=1}^m \mathbb{E}[\|\hat{g}_{z_i}\|_2^2 \mid s] = R^2 \sum_{i=1}^m \left(1 - \|p_i(s)\|_2^2\right), \tag{19}$$

which proves the first equality in (12).

For the upper bound, for any $p \in \Delta^{K-1}$, Cauchy–Schwarz gives $\|p\|_2^2 \geq \frac{1}{K}(\|p\|_1)^2 = \frac{1}{K}$, hence $1 - \|p\|_2^2 \leq 1 - \frac{1}{K}$. Applying this to each $p_i(s)$ yields

$$\mathbb{E}\big[\|\hat{g}_z\|_2^2 \mid s\big] \leq R^2 \sum_{i=1}^m \left(1 - \frac{1}{K}\right) = mR^2\left(1 - \frac{1}{K}\right).$$

Moreover, $\|p\|_2^2 = 1/K$ holds if and only if $p$ is uniform over the $K$ bins, so the inequality is tight if and only if $p_i(s)$ is uniform for all $i$. $\qquad\square$

# B. Experimental Details

## B.1. Hyperparameters

Our implementation is based on the `clean_rl` PPO codebase (Huang et al., 2022b) and the ManiSkill benchmark suite (Tao et al., 2025). Unless otherwise specified, we adopt the default hyperparameters from these codebases without task-specific retuning. For completeness, we report the full set of default parameters in Table 2. For RGB-based tasks, we follow the benchmark suite and use a NatureCNN feature extractor to encode observations into a latent representation, whose dimensionality is given by the extractor's output feature size.

*Table 2.* **Default PPO hyperparameters.** We report the default configurations used for each benchmark family.

| Category | Hyperparameter | Gym | ManiSkill (state) | ManiSkill (RGB) |
|---|---|---|---|---|
| Common | Discount factor $\gamma$ | 0.99 | 0.80 | 0.80 |
| | GAE parameter $\lambda$ | 0.95 | 0.90 | 0.90 |
| | PPO clipping $\epsilon$ | 0.2 | 0.2 | 0.2 |
| | Value loss coefficient | 0.5 | 0.5 | 0.5 |
| | Entropy coefficient | 0 | 0 | 0 |
| | Max grad norm | 0.5 | 0.5 | 0.5 |
| | Num epochs per update | 10 | 4 | 8 |
| | Num minibatches | 64 | 32 | 32 |
| | Advantage normalization | True | True | True |
| | Num envs | 16 | 4096 | 1024 |
| | Num steps | 1024 | 50 | 50 |
| | Total timesteps | 5M | 50M | 50M |
| Optimization | Optimizer | Adam | Adam | Adam |
| | Learning rate | $3 \times 10^{-4}$ | $3 \times 10^{-4}$ | $3 \times 10^{-4}$ |
| | Weight decay | $1 \times 10^{-5}$ | $1 \times 10^{-5}$ | $1 \times 10^{-5}$ |
| Critic Network | Hidden dim | 64 | 256 | 512 |
| | Activation function | Tanh | Tanh | ReLU |
| | Number of Hidden Layers | 1 | 2 | 1 |
| Actor Network (MLP) | Hidden dim | 256 | 256 | 512 |
| | Activation function | Tanh | Tanh | ReLU |
| | Number of Hidden Layers | 2 | 2 | 1 |
| Actor Network (RN) | Hidden dim $d_h$ | 256 | 128 | 512 |
| | Number of Blocks $N$ | 2 | 2 | 1 |

Beyond PPO, we evaluate the same actor–critic architecture with two additional on-policy objectives, Trust Region Policy Optimization (TRPO) and Simple Policy Optimization (SPO), to test whether our actor-design findings transfer beyond the clipped surrogate. Both variants reuse the PPO rollout pipeline, GAE estimation ($\gamma = 0.99$, $\lambda = 0.95$), advantage normalization, and network architectures; **only the policy update is changed**.

For TRPO, we replace the clipped surrogate with a natural-gradient update computed by conjugate gradient using Fisher-vector products from the analytical KL Hessian, followed by backtracking line search to enforce $\delta_{\mathrm{KL}} = 0.01$. We use Fisher damping 0.1, 10 CG iterations, and up to 10 backtracking steps with shrinkage factor 0.8. The actor is updated on the full batch without minibatching or multi-epoch reuse.

For SPO, we keep the same rollout, advantage estimation, and critic update, but replace the policy update with SPO's first-order softly regularized surrogate. All other hyperparameters follow PPO, so performance differences mainly reflect the actor-update rule.

## B.2. Environments

All experiments use the standard benchmark implementations from Gymnasium/MuJoCo and ManiSkill. Unless otherwise specified, we adopt the default environment settings provided by the corresponding libraries and report results as the mean over five random seeds. For the main results, Gymnasium/MuJoCo experiments are run on an NVIDIA H800, while ManiSkill experiments are run on 2× NVIDIA GeForce RTX 5090 GPUs. All TRPO and SPO experiments are run on eight RTX 6000 Pro GPUs.

*Table 3.* **TD3-normalization reference scores** used for Gym locomotion aggregation.

| Environment | Random | TD3 |
|---|---|---|
| Ant-v4 | -70.288 | 3942 |
| HalfCheetah-v4 | -289.415 | 10574 |
| Hopper-v4 | 18.791 | 3226 |
| Humanoid-v4 | 120.423 | 5165 |
| Walker2d-v4 | 2.791 | 3946 |

**Gym – Locomotion.** We evaluate on five widely-used MuJoCo locomotion tasks from Gymnasium (`-v4` versions): `HalfCheetah-v4`, `Ant-v4`, `Hopper-v4`, `Humanoid-v4`, and `Walker2d-v4`. We follow standard practice and do not apply additional preprocessing beyond the default observation and action interfaces of the environments. When aggregating scores across tasks, we report *normalized return* using TD3-normalization,

$$\text{TD3-Normalized}(x) := \frac{x - x_{\text{random}}}{x_{\text{TD3}} - x_{\text{random}}}, \tag{20}$$

where $x$ denotes the raw episodic return, and $(x_{\text{random}}, x_{\text{TD3}})$ are the random-policy and TD3 reference scores, respectively (Table 3).

**ManiSkill (state).** We additionally benchmark on state-based manipulation tasks from ManiSkill (Tao et al., 2025). Each environment provides low-dimensional proprioceptive/state observations, and we evaluate performance using the standard ManiSkill success metric (success rate). We use the default task configurations and consider 10 tasks. Across tasks, we keep the default algorithmic hyperparameters fixed and only vary throughput-related settings (e.g., number of parallel environments and rollout length). All runs use a total budget of 50M environment steps per task.

**ManiSkill (RGB).** For vision-based control, we evaluate on five ManiSkill tasks with RGB observations. We use the default RGB observation interface provided by ManiSkill and report success rate. Similar to the state-based setting, we keep the algorithmic hyperparameters fixed and only adjust throughput-related settings for stable training. All runs use 50M environment steps per task.

## C. Extended Study

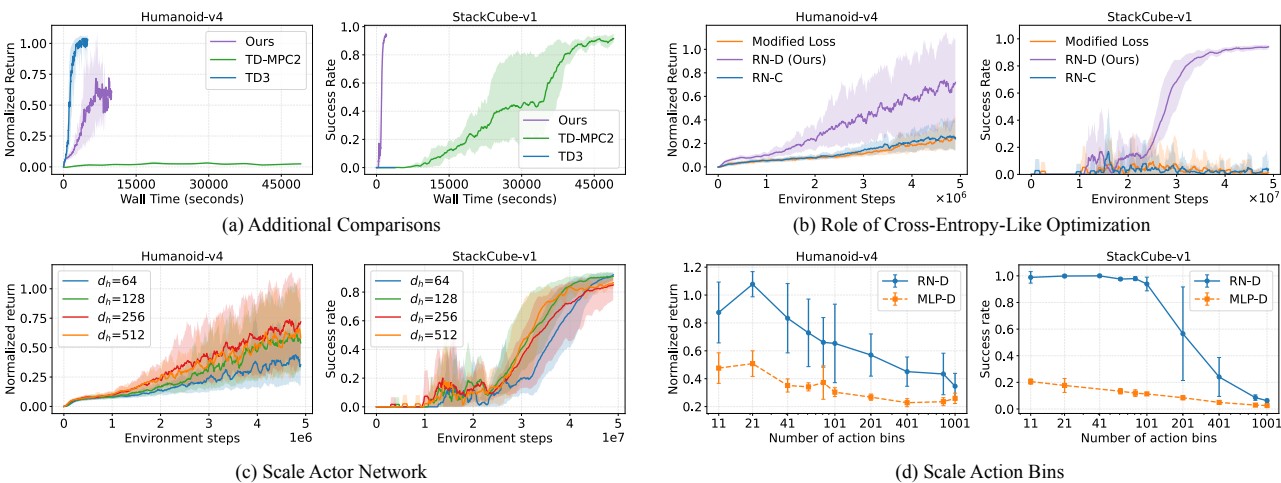

*Figure 8.* Extended studies on sample efficiency, optimization objective, and scaling.

### C.1. Additional Comparisons with State-of-the-Art Methods

To complement our main evaluation, we compare against two recent state-of-the-art continuous-control methods: TD3 (Fujimoto et al., 2018) and TD-MPC2 (Hansen et al., 2024). Since these methods differ in their interaction patterns and training

pipelines, we report wall-clock time curves. Our method and TD3 are run on the same hardware configuration ($2\times$ RTX 5090 GPUs), while TD-MPC2 is run on an NVIDIA A200 GPU, which is typically faster, and this makes the wall-clock comparison conservative with respect to TD-MPC2. We use the authors' published packages/official implementations for TD3 and TD-MPC2, and follow their standard training protocols. TD-MPC2 are evaluated on the same tasks with 3 random seeds (Hansen et al., 2024).

Figure 8 (a) compares wall-clock learning curves on Humanoid-v4 and StackCube-v1. On Humanoid-v4, TD3 improves quickly early on, reaching high normalized return within a short time window, whereas TD-MPC2 makes little progress under the same horizon. Our approach is on-policy and built on PPO, it achieves substantial gains and outperforms standard PPO baselines, continuing to improve with more interaction. On StackCube-v1, our method rapidly reaches high success rates, while TD3 fails to learn and TD-MPC2 only improves much later in training. Overall, these results highlight that the proposed on-policy discretized actor is particularly effective for challenging manipulation tasks where exploration and optimization are difficult, and that it remains competitive in locomotion while offering favorable wall-clock efficiency.

### C.2. Ablation study on the role of cross-entropy-like optimization

Discretized categorical policies differ from Gaussian policies in two tightly coupled aspects: (i) the discretized action representation induced by binning, and (ii) the likelihood model used for optimization, which yields a cross-entropy-like objective rather than a squared-error objective. To disentangle these effects, we introduce a *loss-swap* ablation that isolates the role of the optimization objective while holding the action representation space fixed.

Starting from our discrete categorical policy, we modify only the likelihood used for policy optimization. The actor still outputs logits over $K$ bins for each action dimension, inducing probabilities $p_{\theta,i}(k \mid s) = \text{softmax}(z_{\theta,i}(s))_k$. Instead of sampling from the categorical distribution, we compute the expected action as the probability-weighted average of bin centers $\{c_k\}_{k=1}^K$,

$$\mu_{\theta,i}(s) \;=\; \sum_{k=1}^K p_{\theta,i}(k \mid s)\, c_k, \tag{21}$$

and treat $\mu_\theta(s)$ as the mean of a Gaussian policy,

$$\pi_\theta^{\text{swap}}(a \mid s) \;=\; \mathcal{N}\big(a; \mu_\theta(s), \Sigma\big). \tag{22}$$

Policy updates are then performed using the standard Gaussian log-likelihood,

$$\log \pi_\theta^{\text{swap}}(a \mid s) = -\tfrac{1}{2}(a - \mu_\theta(s))^\top \Sigma^{-1}(a - \mu_\theta(s)) + \text{const}, \tag{23}$$

which is equivalent to a weighted squared-error objective under fixed $\Sigma$. This loss-swapped variant uses the same network architecture and discretized action representation as the original categorical policy, differing only in the likelihood (and thus the optimization loss) used for policy updates.

Figure 8 (b) compares RN-D, the loss-swapped variant, and a continuous Gaussian baseline (RN-C). Replacing the categorical likelihood with a Gaussian likelihood leads to a substantial performance drop, with the loss-swapped variant closely matching the continuous Gaussian baseline. These results show that discretizing the action space alone is insufficient to explain the observed gains. Even with a discretized representation, switching from a cross-entropy-like objective to a squared-error objective eliminates most of the performance improvement, highlighting the central role of the categorical likelihood and its induced optimization geometry.

### C.3. Ablation study on the capacity of the actor network

To study how actor capacity affects on-policy learning with discretized categorical policies, we scale the actor network by varying the hidden width $d_h \in \{64, 128, 256, 512\}$. Note that the maximum hidden dimension could be $4d_h$. We keep the rest of the training pipeline fixed, including the PPO objective, the critic architecture, and all other hyperparameters, so that differences can be attributed to actor capacity.

Figure 8(c) shows that scaling actor width generally improves optimization speed on both tasks. On Humanoid-v4, wider actors learn faster and attain higher returns during much of training, with the largest gains when increasing from $d_h = 64$ to $d_h \in \{128, 256\}$. However, the benefits do not always translate to higher asymptotic performance, and the widest setting ($d_h = 512$) can yield similar or even slightly lower final returns. On StackCube-v1, larger actors typically reach high success

earlier, while final success rates are comparable. Overall, actor capacity is an effective lever for accelerating on-policy learning, but its impact on final performance can be non-monotonic.

### C.4. Ablation study on the discrete bins

Another factor affecting the performance of discrete policies is the number of action bins, which controls the granularity of the discretized action space. Fewer bins lead to a coarser representation, while more bins provide finer resolution and more closely approximate continuous control. To examine this effect, we sweep the number of action bins from 21 to 1001, with 41 bins used in the main experiments.

Figure 8 shows how policy performance varies with the number of action bins. For MLP-based discrete policies without residual structure (MLP-D), performance generally degrades as the number of bins increases, but the degradation is relatively mild. When the number of bins is smaller than the hidden dimension, MLP-D performs better than or is at least comparable to the continuous baseline (MLP-C). On StackCube-v1, the success rate decreases almost linearly with the logarithm of the number of action bins (the x-axis is log-scaled), even though the size of the factorized action space grows linearly with the number of bins. This indicates that the performance gains of discrete policies are not solely due to finer action resolution, but are also influenced by optimization behavior.

For RN-D methods, discrete policies achieve strong performance with a small number of bins and consistently outperform the off-policy TD3 baseline on Humanoid. As the number of bins increases, the mean return decreases. On Humanoid, although the average performance drops, the variance across five random seeds remains relatively stable when increasing the number of bins from 11 to 101. A similar trend appears on StackCube-v1, where performance is largely preserved for small to moderate bin counts, followed by a sharp decline once the number of bins exceeds 100.

Overall, these results show that discrete policy performance is sensitive to discretization granularity. With relatively few bins, the optimization advantages introduced by discretization—such as a simpler optimization landscape—outweigh the loss in action resolution. However, as the number of bins becomes large, the policy must predict increasingly fine-grained action distributions, which likely exceeds the effective representational or optimization capacity of the network, leading to the observed performance degradation. A promising way to mitigate this issue is to introduce more efficient action tokenization schemes, which aim to preserve high action precision while retaining the optimization advantages of discrete policies.

### C.5. Additional Runtime Analysis

We provide additional runtime analyses to examine the computational overhead of the discretized actor parameterization. In particular, we study two aspects: the effect of the number of discretization bins $K$ on training throughput, and the wall-clock training time of different actor architectures.

*Table 4.* Effect of bin count $K$ on training throughput, measured by samples per second (SPS), and wall-clock time on Humanoid-v4. The action dimension is 17. Results are averaged over 5 seeds.

| $K$ (bins) | 11 | 21 | 41 | 61 | 101 | 201 | 401 | 801 | 1001 |
|---|---|---|---|---|---|---|---|---|---|
| Avg SPS | 1789 | 1342 | 974 | 720 | 721 | 724 | 719 | 716 | 737 |
| Time (h) | 0.76 | 1.27 | 1.74 | 1.95 | 1.96 | 1.96 | 1.97 | 1.98 | 1.95 |

Table 4 shows that increasing the number of bins initially reduces throughput, as expected, since a larger categorical action head introduces additional computation. However, the runtime overhead saturates once $K$ becomes moderately large. In particular, after $K = 61$, both SPS and wall-clock time remain nearly unchanged, suggesting that the training pipeline is no longer dominated by the cost of increasing the categorical output dimension. This indicates that using a sufficiently fine discretization does not introduce prohibitive computational overhead in practice.

Table 5 further compares the runtime of continuous and discretized actor variants with both MLP and residual-network backbones. The discretized variants do not incur additional wall-clock overhead compared with their continuous counterparts. In fact, MLP-D and RN-D achieve slightly higher SPS and comparable or lower wall-clock time than MLP-C and RN-C, respectively. These results suggest that the proposed discretized actor parameterization improves policy representation without sacrificing training efficiency.

*Table 5.* Wall-clock time and throughput, measured by samples per second (SPS), averaged across 5 Gym locomotion tasks with 5 seeds each.

| Method | Wall-clock Time (h) | SPS |
| --- | --- | --- |
| MLP-C | 1.65 | 854.4 |
| MLP-D | 1.56 | 925.3 |
| RN-C | 1.75 | 811.1 |
| RN-D | 1.64 | 873.8 |

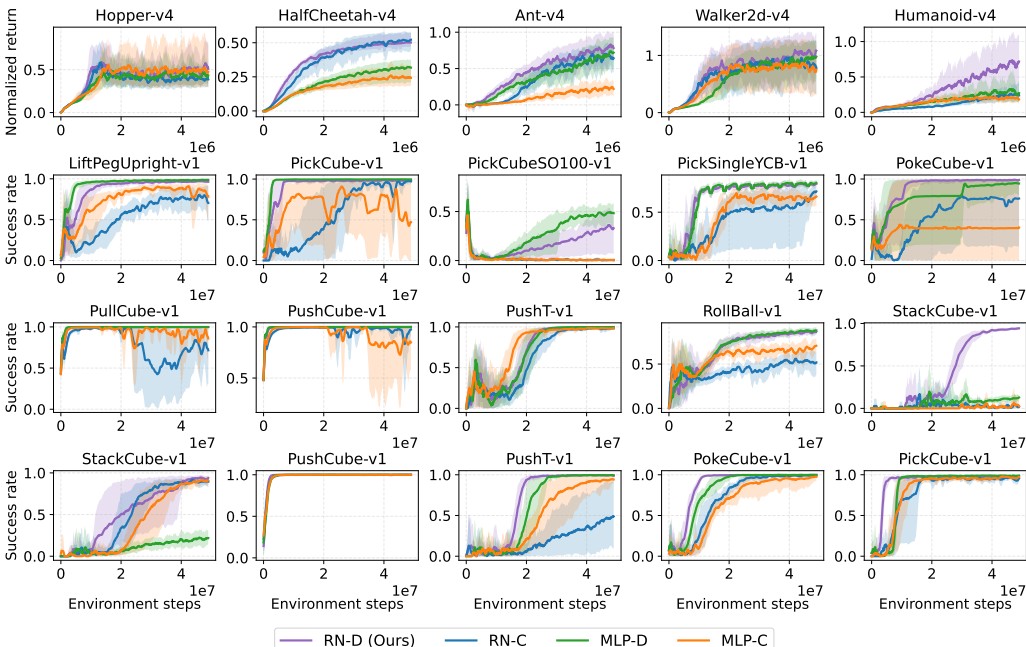

*Figure 9.* Complete learning curves across all tasks under PPO algorithm. We report normalized return on MuJoCo benchmarks (top row) and success rate on ManiSkill manipulation benchmarks (remaining rows), including both state-based and RGB-based variants. Curves are averaged over 5 random seeds; shaded regions denote 95% stratified bootstrap confidence intervals.

## D. Complete main results

We provide the complete learning curves for all evaluated tasks. Vision-based tasks are evaluated only under PPO, since PPO is the dominant and most widely used on-policy algorithm among the three considered objectives. In addition, TRPO has substantially higher memory requirements in our implementation and cannot use the same number of parallel environments under the same hyperparameter setting for vision-based tasks. The figure includes per-task performance over environment steps—normalized return for MuJoCo benchmarks and success rate for ManiSkill manipulation benchmarks, including both state-based and RGB-based settings where applicable. Curves show the mean across five random seeds with shaded regions indicating variability, and are included here for completeness and transparency beyond the aggregated main-text results.

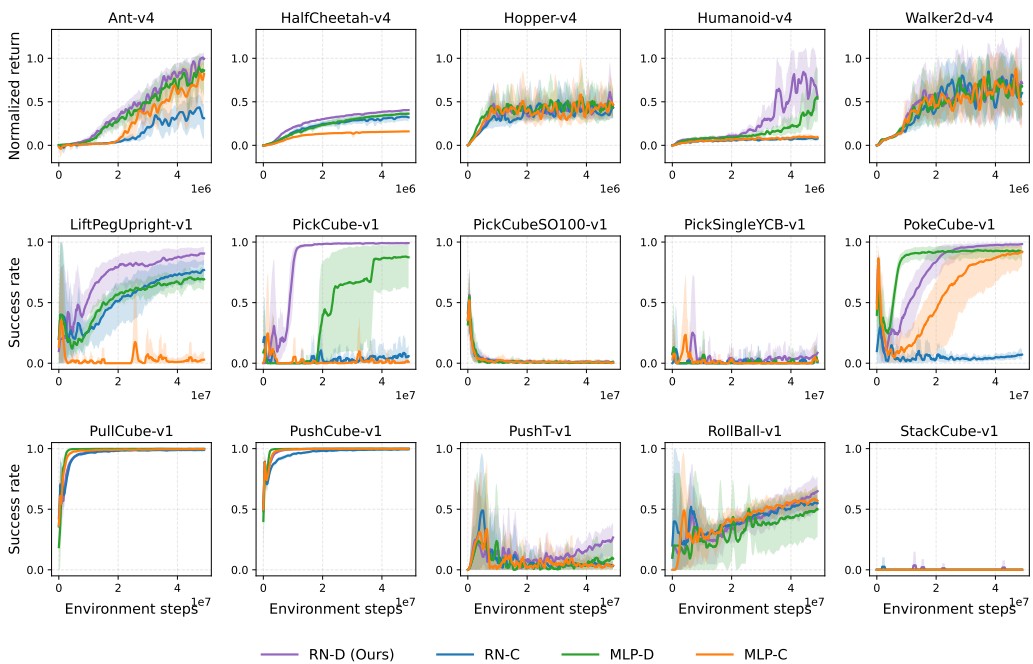

*Figure 10.* Complete learning curves across all tasks under TRPO algorithm. We report normalized return on MuJoCo benchmarks (top row) and success rate on ManiSkill manipulation benchmarks (remaining rows), including both state-based and RGB-based variants. Curves are averaged over 5 random seeds; shaded regions denote 95% stratified bootstrap confidence intervals.

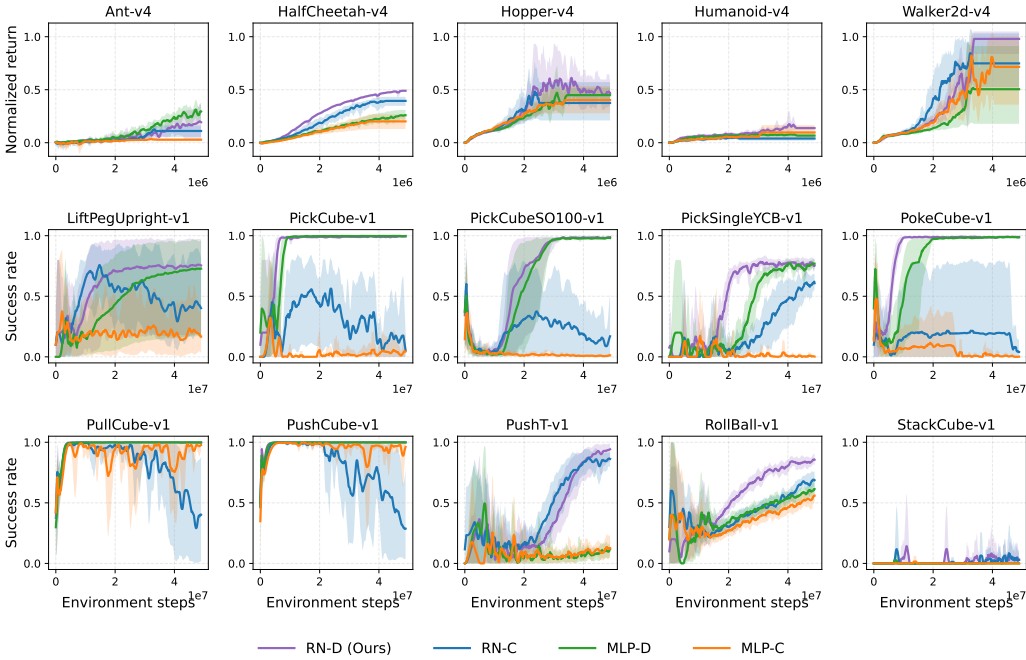

*Figure 11.* Complete learning curves across all tasks under SPO algorithm. We report normalized return on MuJoCo benchmarks (top row) and success rate on ManiSkill manipulation benchmarks (remaining rows), including both state-based and RGB-based variants. Curves are averaged over 5 random seeds; shaded regions denote 95% stratified bootstrap confidence intervals.

