# OpenReview forum: "RN-D: Discretized Categorical Actors for On-Policy Reinforcement Learning"
_ICML.cc/2026/Conference — ICML 2026 regular_

### Official Review · Reviewer_nbkF · 2026-03-10

**Soundness:** 3
**Presentation:** 3
**Significance:** 2
**Originality:** 3
**Overall Recommendation:** 4
**Confidence:** 4

**Summary:**

This paper finds that changing the actor parameterization and actor network design could enhance performance, and proposes RN-D, a variant of PPO that replaces the standard diagonal-Gaussian actor with a factorized discretized categorical actor over action bins, and pairs it with a pre-LayerNorm residual actor network.

**Compliance With Llm Reviewing Policy:**

Affirmed.

**Final Justification:**

My major concern is that the contribution still seems best framed as a strong actor-design recipe for on-policy RL rather than a generally superior continuous-control technique, and the broader significance claim should remain somewhat narrower.

And the authors clarified the paper’s intended scope, added supporting results beyond PPO, and made it clear that the contribution should be understood as an actor-design improvement for on-policy methods.

Hence, my final recommendation is weak accept.

**Key Questions For Authors:**

If the trick is general and the authors provide new experiments to support it, I will reconsider the rating.
1. What is  the extra computational does your RN-D introduce, compared to standard PPO？
2. Could the authors clarify whether the claim is “a better PPO actor design,” or a generally superior continuous-control technique? Better to attach experiment results to support this.
3. In Figure 3, in the Gym locomotion row, why are all the values below 1? Does that mean all the variants could not outperform even TD3, not to mention other advanced algos? Is the gap mainly due to the PPO backbone? If you apply your trick to TD3, will it also improve performance? Or can this trick only apply to on-policy algos, but not off-policy algos?
4. Why choose to fix the critic design? Does this also limit the results, or is it an advantage? Why not consider a joint design?
5. The related work discusses several closely related algos,why these methods are not included in the experimental comparison?

**Limitations:**

* the method is only validated under PPO, but the title is "for On-Policy Reinforcement Learning". There exists mismatch.

**Strengths And Weaknesses:**

**Strengths**
- The writing is clear, and the authors explicitly claim that the contribution is an actor-side change.
- The loss-swap ablation offers some insight: the categorical likelihood / cross-entropy-like optimization structure also contributes to the performance.


**Weaknesses**
1. Figure 3 only compares four PPO variants, and does not compare to other advanced algos. Also, the performance of all the variants lies behind TD3. I think this supports a narrower claim.
2. About novelty, discretized actors are not new, and residual / normalized MLP are also not new. The contribution, together with the experiments, seems to propose a trick that is well suited for PPO. Although this trick might be extended to other backbone algos, no experiments support it. I acknowledge that a general, simple, and effective trick is important, especially if it is general rather than tailored to one method. Thus, I am not fully convinced by the claimed value beyond a PPO implementation trick.
3. The authors choose to keep the critic design unchanged. I do not think it quite makes sense to just change the actor side without considering the actor-critic interaction.

---

> ### Author Rebuttal · Authors · 2026-03-30
>
> We thank the reviewer for the thoughtful and constructive feedback. Below we clarify the reviewer's concerns.
>
> **(1) Generality beyond PPO (W1, W2, Q2, L1).**
> We thank the reviewer for raising this important point regarding generality beyond PPO. While our main experiments focus on PPO to enable controlled comparisons of actor parameterizations, the proposed discretized actor is **algorithm-agnostic** and can be directly applied to other **on-policy policy-gradient methods**.
>
> To support this, we extend our evaluation to Trust Region Policy Optimization (TRPO; Schulman et al., 2015) and Simple Policy Optimization (SPO; Liu et al., 2024), comparing four variants (RN-D, MLP-D, RN-C, MLP-C) on both Gym (5 tasks) and ManiSkill (10 tasks). As shown in [Figure 4](https://anonymous.4open.science/r/Rebuttal-figure-table-ICML26-BA46/rebuttal_only_figure.md) and [Table 1](https://anonymous.4open.science/r/Rebuttal-figure-table-ICML26-BA46/rebuttal_only_figure.md), the same trend consistently holds across both algorithms. In particular, RN-D achieves the best performance throughout training, with clear improvements in both sample efficiency and final performance.
>
> These results demonstrate that the effectiveness of our approach is not specific to PPO, but instead stems from improving the stochastic policy parameterization, which generalizes across on-policy RL algorithms. We will include these results in the revised version to better highlight the generality of our method.
>
> **(2) Performance of Gym Locomotion and Compatibility with off-policy algorithms (Q3).**  PPO can underperform strong off-policy methods on gym benchmarks, and this is a well-known gap in the literature and not specific to our method. Our goal is to study relative improvements within on-policy RL, rather than to outperform all off-policy algorithms. That said, we observe that our approach consistently achieves strong performance on ManiSkill tasks, and in [Figure 2](https://anonymous.4open.science/r/Rebuttal-figure-table-ICML26-BA46/rebuttal_only_figure.md) (with longer training horizons), it can even surpass TD3, suggesting that improved optimization can help narrow this gap.
>
> Regarding applicability to off-policy methods, our approach is specifically designed to improve stochastic policy-gradient estimators used in on-policy algorithms, where the actor is optimized via $\nabla_\theta \log \pi_\theta(a|s)$. In this setting, replacing a diagonal Gaussian actor with a discretized categorical actor directly modifies the policy-gradient estimator, leading to lower gradient variance and more stable optimization (Sec. 4.1).
> In contrast, deterministic policy gradient methods such as TD3 and DDPG use
> $\nabla_\theta J(\theta) = \mathbb{E}\_{s\sim \mathcal{D}} \big[ \nabla_a Q_\phi(s,a)\mid_{a=\mu_\theta(s)} \nabla_\theta \mu_\theta(s) \big],$
> which does not involve $\nabla_\theta \log \pi_\theta(a|s)$ or stochastic action sampling.
> Instead, the dominant variance arises from the critic—errors in estimating $Q_\phi$ and $\nabla_a Q_\phi$. Therefore, the variance-reduction benefits of RND do not directly transfer to off-policy algorithms.
>
> We will clarify this distinction and the evaluation scope in the final version to avoid potential confusion.
>
>
> **(3) Actor–critic interaction and fixing the critic (W3, Q4)**.
> We thank the reviewer for this insightful comment. Our decision to keep the critic unchanged is intentional, as it enables us to isolate and clearly evaluate the impact of policy parameterization on on-policy optimization. This controlled design follows established empirical practice, for example, *What Matters In On-Policy Reinforcement Learning? A Large-Scale Empirical Study* evaluates over 25,000 agents across Gym tasks by varying one component(e.g., network architecture, activation function, optimizer) at a time while keeping others fixed. Notably, that study does not find the policy network to be a significant factor; however, all policies considered are based on Gaussian parameterizations. In contrast, our results suggest that the choice of policy distribution itself can fundamentally affect optimization, revealing a dimension that is not captured within Gaussian-only design spaces.
>
> **(4) Computational overhead (Q1).**
> The residual network in RN-D introduces only a modest computational overhead. We benchmark wall-clock time and samples per second (SPS) averaged across the gym benchmarks in [Table 2](https://anonymous.4open.science/r/Rebuttal-figure-table-ICML26-BA46/rebuttal_only_figure.md).
> Compared to standard PPO, our proposed RN-D achieves comparable wall-clock time (1.64 h vs. 1.65 h) and even slightly higher throughput. Overall, the computational cost of RN-D is negligible relative to the significant performance gains reported in our main results. We will include these computational results in the final version.
>
> We hope these results address the reviewer’s concern and support reconsideration of the rating.

---

> > ### Author Rebuttal · Reviewer_nbkF · 2026-04-02
> >
> > Thank the authors for the rebuttal. However, I do not think it fully resolves my main concern: the contribution still seems best framed as a strong actor-design recipe for on-policy RL rather than a generally superior continuous-control technique, and the broader significance claim should remain somewhat narrower.

---

> > > ### Author Response · Authors · 2026-04-02
> > >
> > > Thank you very much for your follow-up comments. We greatly appreciate the opportunity to clarify the scope and contributions of our work.
> > >
> > > Our core contribution is to revisit policy representation as a deliberate design choice for on-policy reinforcement learning. Specifically, we demonstrate that replacing the standard Gaussian actor with a discretized categorical actor, paired with regularized networks, yields consistent and substantial performance improvements within the on-policy regime. This focus is explicitly reflected in our title ("for On-Policy Reinforcement Learning") and is emphasized throughout the paper.
> > >
> > > To directly address your original concern that our method was only validated under PPO, we have extended our evaluation to two additional on-policy algorithms, TRPO and SPO. Across four actor variants and a diverse set of environments, our results consistently show that the proposed RN-D variant achieves the best performance across all three on-policy algorithms, in terms of both sample efficiency and final return.
> > >
> > > We fully understand your current concern that our contribution might be misinterpreted as a generally superior technique for all continuous control settings. We completely agree with this distinction, and we wish to clarify that this is not a claim we intend to make. Our method targets improvements to the stochastic policy gradient estimator, which is specific to on-policy methods and structurally absent in deterministic policy gradient algorithms such as TD3. In the final revised version, we will carefully revise the writing throughout the paper to ensure this scope is communicated clearly and consistently, eliminating any potential for unintended broader interpretations.
> > >
> > > Thank you again for your thoughtful feedback, and we hope these clarifications address your concerns fully.

---

### Official Review · Reviewer_E9c5 · 2026-03-12

**Soundness:** 3
**Presentation:** 3
**Significance:** 3
**Originality:** 3
**Overall Recommendation:** 5
**Confidence:** 4

**Summary:**

As standard deep reinforcement learning implementations for continuous control tasks typically rely on Gaussian actors and relatively small MLPs, this paper revisits that policy choice and studies discretized categorial actors that represent each action dimension with a distribution over bins. That is, instead of the usual multivariate Gaussian parameterization, the paper proposes instead to use a factorized categorical policy, turning the policy loss into a cross-entropy like objective over the selected bins. When paired with regularized actor networks, numerical experiments on standard deep reinforcement learning benchmarks show that this parameterization usually outperforms PPO both in terms of final reward and better sample efficiency.

**Compliance With Llm Reviewing Policy:**

Affirmed.

**Final Justification:**

The additional experiments strengthen the paper. In particular, the PPO-CMA comparison addresses my concern about the strength of the original baselines, the added TRPO/SPO results make the empirical claim broader, and the longer-horizon experiments suggest the improvement is not only a sample-efficiency effect. The added wall-clock measurements also alleviate my concern that the discretized output expansion could introduce a meaningful computational overhead.

My remaining reservation is mainly about scope and interpretation. I still believe the theoretical discussion should be presented more as intuition than as a thorough explanation of the practical training dynamics, and the empirical claims should be calibrated accordingly. Overall, the rebuttal addressed most of my original concerns, and so I am updating my score accordingly.

**Key Questions For Authors:**

1. Does the output expansion (action dimension vs. bins) add any training overhead?

2. Do Gaussian baselines remain significantly worse under longer training, or is the main advantage of the proposed method primarily sample efficiency?

**Limitations:**

Yes.

**Strengths And Weaknesses:**

Strengths:
1. Good empirical evaluation, as the proposed method is tested across locomotion, state-based manipulation, and vision-based manipulation tasks. Reported gains on Humanoid-v4, StackCube-v1, in particular, are pretty substantive.
2. The numerical experiments (both in terms of the proposed end-to-end framework, and on the impact of regularized networks on Gaussian policies) and the discussion in the paper clearly motivate that policy parameterization matters for on-policy learning deep RL.

Weaknesses:
1. The provided experiments only compare the proposed parameterization against PPO with a multivariate Gaussian policy. It would be interesting to also see comparisons against PPO-CMA, since that algorithm aims to reduce variance shrinkage in PPO, and to more deep RL algorithms in the main paper.
2. The main theoretical result (Proposition 4.1) studies a fairly idealized setting, with a one-step REINFORCE estimator at a fixed state and with a constant return, and it is not clear to me that that theoretical analysis carries over to more general cases. Moreover, for the Gaussian parameterization, the analysis assumes that the covariance is fixed. But in practice the variance is often learned as well.

---

> ### Author Rebuttal · Authors · 2026-03-30
>
> We sincerely thank the reviewer for the constructive feedback and encouraging comments. Below, we address the questions on baseline comparisons, the scope of the theoretical analysis, computational overhead, and long-horizon training behavior.
>
> **(1) Comparison to PPO-CMA and additional baselines (W1)**
>
> We thank the reviewer for this helpful suggestion. In the revised version, we include PPO-CMA as an additional baseline, which is specifically designed to mitigate variance shrinkage in PPO. Results are in [Figure 3](https://anonymous.4open.science/r/Rebuttal-figure-table-ICML26-BA46/rebuttal_only_figure.md).
> Beyond PPO-based comparisons, we also extend our evaluation to other on-policy algorithms, including Trust Region Policy Optimization (TRPO; Schulman et al., 2015) and Simple Policy Optimization (SPO; Liu et al., 2024). We compare four variants (RN-D, MLP-D, RN-C, MLP-C) across both Gym (5 tasks) and ManiSkill (10 tasks).
> Results are in [Figure 4](https://anonymous.4open.science/r/Rebuttal-figure-table-ICML26-BA46/rebuttal_only_figure.md).
> Across all settings, we observe our proposed RN-D achieves the best performance in terms of both sample efficiency and final returns/success rates. These results demonstrate that our approach is not limited to PPO, and that the benefits of improved policy parameterization extend across multiple on-policy RL algorithms.
>
> We will include these additional baselines and experiments in the revised paper to strengthen the empirical evaluation.
>
> **(2) Scope of the theoretical analysis (W2)**
> We appreciate the reviewer’s observation that Proposition 4.1 studies a simplified setting.
> The goal of this analysis is to provide intuition on gradient variance and conditioning differences induced by policy parameterization, rather than a full convergence guarantee.
> We agree that the theorem is derived under a simplified one-step REINFORCE setting; however, it illustrates how gradient variance depends on the policy standard deviation, regardless of how that standard deviation is obtained. Even when the standard deviation is learned in practice, the same qualitative behavior holds: for Gaussian policies, gradient magnitude scales inversely with the standard deviation, and the learned standard deviation typically decreases during training, which amplifies gradient variance. Importantly, our empirical results show that these qualitative insights extend to practical multi-step PPO training, where Gaussian policies exhibit significantly higher and increasing gradient variance.
>
> **(3) Computational overhead from output expansion (Q1)**
> We investigate the computational overhead of RND empirically along two axes: (a) varying the number of bins $K$, and (b) comparing across actor variants.
>
> **Effect of bin count.** We benchmark wall-clock time and SPS on Humanoid-v4 ($d{=}17$) across a wide range of $K$ in [Table 2](https://anonymous.4open.science/r/Rebuttal-figure-table-ICML26-BA46/rebuttal_only_figure.md).
>
> We observe increasing bins from 41 to 1001 adds virtually no overhead. This is because the computational bottleneck in on-policy RL lies in environment simulation and rollout collection, not in the policy forward/backward pass. The output expansion cost is negligible relative to the overall training pipeline.
>
> **Effect of architecture.** We also benchmark across four actor variants on Gym in [Table 3](https://anonymous.4open.science/r/Rebuttal-figure-table-ICML26-BA46/rebuttal_only_figure.md).
> Compared to standard PPO (MLP-C), our proposed RN-D achieves comparable wall-clock time (1.64 h vs. 1.65 h) and even slightly higher throughput. The residual architecture adds ~5–6% overhead over MLP, but the discretized output itself does not increase cost — in fact, MLP-D is faster than MLP-C, likely because categorical sampling is cheaper than Gaussian reparameterization.
>
> **(4) Long-horizon performance vs. sample efficiency (Q2)**
> We thank the reviewer for this important question. To investigate this, we extend training to substantially longer horizons (4× longer than in the main results) on both Humanoid-v4 and StackCube-v1. As shown in [Figure 2](https://anonymous.4open.science/r/Rebuttal-figure-table-ICML26-BA46/rebuttal_only_figure.md), the Gaussian baseline (standard PPO) does not catch up with longer training. On Humanoid-v4, it plateaus early and remains significantly below our method. On StackCube-v1, it fails to achieve meaningful success, while our method quickly reaches and maintains near-optimal performance.
> This gap is consistent with the observed optimization behavior. The Gaussian policy exhibits persistently high and unstable policy gradient variance, whereas our method maintains near-zero variance throughout training.
>
> These results suggest that the advantage of our approach is not limited to improved sample efficiency: high gradient variance in standard PPO can obscure useful learning signals and hinder optimization, ultimately leading to inferior RL performance.

---

> > ### Author Rebuttal · Reviewer_E9c5 · 2026-04-03
> >
> > Thank you for the detailed rebuttal. The additional experiments strengthen the paper.
> >
> > In particular, the PPO-CMA comparison addresses my concern about the original baselines, and the added TRPO/SPO results make the empirical claim broader. The longer-horizon experiments are also useful, since they suggest the improvement is not only a sample-efficiency effect. Finally, the added wall-clock measurements alleviate my concern that the discretized output expansion could introduce a meaningful computational cost.
> >
> > One result I found especially interesting is the large gap between RN-D and PPO-CMA on StackCube. Since this task seems to benefit much more than Humanoid, some discussion of why the advantage is especially large there would strengthen the paper. It would also be interesting to compare PPO-CMA, RN-D, and TD3 more directly on those tasks.
> >
> > My remaining reservation is mainly about scope and interpretation. I still believe the theoretical discussion should be presented more as intuition rather than as a thorough explanation of the training dynamics. I also believe the paper should calibrate its empirical claims carefully.
> >
> > Overall, the rebuttal addressed most of my original concerns, and I am updating my score accordingly.

---

> > > ### Author Response · Authors · 2026-04-04
> > >
> > > We thank the reviewer for the positive assessment and constructive suggestions.
> > >
> > > **On the RN-D vs. PPO-CMA gap on StackCube.**
> > > StackCube requires precise, multi-phase manipulation (grasp, lift, place, stabilize), making optimization particularly sensitive to error accumulation. In such settings, noisy or unstable gradients can compound over time and significantly degrade performance. Our method improves gradient signal quality and stability, which we believe contributes to the observed performance gap. While PPO-CMA enhances exploration via covariance adaptation, it remains within the Gaussian parameterization and does not benefit from the same stability properties. In contrast, locomotion tasks such as Humanoid involve smoother dynamics and are less sensitive to these effects, leading to smaller but still consistent gains. We will clarify this distinction and include a direct PPO-CMA vs. RN-D vs. TD3 comparison in the revision, as suggested.
> > >
> > > **On theoretical framing.**
> > > We agree and will present Proposition 4.1 explicitly as an illustrative intuition for gradient variance behavior, rather than as a general theoretical guarantee.
> > >
> > > **On calibrating empirical claims.**
> > > We fully agree and will revise the paper to clearly scope our claims to the on-policy setting and avoid broad SOTA statements. We will update the abstract and main text to ensure precise and accurate positioning.

---

### Official Review · Reviewer_1E8Z · 2026-03-13

**Soundness:** 3
**Presentation:** 4
**Significance:** 2
**Originality:** 2
**Overall Recommendation:** 5
**Confidence:** 4

**Summary:**

This paper provides a systematic study on replacing the Gaussian MLP policy with a categorical, discretized policy with residual networks. It first reviews the connections between policy gradients of Gaussian/categorical policies and different types of supervised learning losses (regression/cross-entropy). It also revisits the gradient variance of Gaussian/categorical and empirically validates it. To further enhance policy learning, the paper investigates the use of residual structure in policy networks. Evaluation in two continuous control benchmarks shows significant improvement when switching to the categorical discretized policy with residual networks.

**Compliance With Llm Reviewing Policy:**

Affirmed.

**Final Justification:**

I increased my rating from 4 to 5 as my concerns are addressed. I also checked other reviews and responses and didn't find any issues significant to me.

**Key Questions For Authors:**

Addressing the following questions may strengthen the empirical contributions of the paper:

1. Have the authors considered $K=2$ and $K=3$? It’d be interesting to expand the range to these values to check whether an intermediate value is best.
2. What are the implications of the signal-to-noise ratio analysis? Could the authors contextualize this study to make the message clearer?

Questions related to the interpretations of the current empirical results:

3. Why are there upward tilts in the gradient variance curves of discrete policies in Figure 4? What happens after step $5e6$?
4. How are the hyperparameters chosen? Why is ManiSkill-RGB easier than ManiSkill-State for some algorithm variants in StackCube-v1?

**Limitations:**

The paper should discuss potential limitations of applying discrete policies, e.g., in the high dimensional settings.

**Strengths And Weaknesses:**

The paper has several notable strengths, including its soundness, significance, and presentation.
1. The paper delivers a rigorous, systematic study on the design of PPO’s policy parameterization and network structure. The two dimensions are studied jointly, using reasonable experimental protocols and yielding statistically significant results.
2. The design of the policy’s parameterization and network structure is fundamental in empirical RL, and the paper provides clear insights into these choices, including the benefits of categorical parameterization and residual networks with LayerNorm.
3. The presentation of the paper is also excellent with clear structure, diagrams, and figures.

On the other hand, it also has weaknesses in its originality and in some aspects of significance.
1. While the paper does investigate the impact of the number of action bins $K$ across a wide range of larger values, the range does not cover small values below $11$. This is a limitation as the current results indicate that the smallest value investigated ($K=11$) is not statistically significantly worse than a higher value used, leaving a gap in this hyperparameter study.
2. The approach investigated in the paper is not new, and the quantity and significance of the novel insights it provides are limited.

Minor points/suggestions
1. Why is the approach call *regularized* networks?
2. In Line 076R, it is not clear what CMA-ES is, which could confuse the reader.

---

> ### Author Rebuttal · Authors · 2026-03-30
>
> We sincerely thank the reviewer for the careful reading. We address each concern below.
>
> **(1) Discretization with small K (K = 2, 3, 5)**
>
> We have run the requested experiments on Humanoid-v4 and StackCube-v1 with 5 seeds each.
> The results in [Figure 1](https://anonymous.4open.science/r/Rebuttal-figure-table-ICML26-BA46/rebuttal_only_figure.md) reveal a task-dependent trade-off: on Humanoid-v4, even bin=2 outperforms MLP-C and bin=3/5 already match bin=41; on StackCube-v1, finer discretization is needed, with smaller bins showing early plateaus and higher variance. Overall, bin selection trades off computational efficiency against action granularity. We will include this ablation in the revision.
>
> **(2) Novelty and significance**
>
> While individual components such as discretization and residual network architectures are not new in isolation, our key contribution is a unifying insight: policy parameterization fundamentally alters the underlying optimization objective. In particular, Gaussian policies correspond to a weighted mean-squared-error objective, whereas discretized policies induce a weighted cross-entropy objective. This shift leads to substantially improved optimization stability and empirical performance.
> We further emphasize that prior large-scale empirical studies (e.g., What Matters in On-Policy RL?, which evaluates over 25,000 agents on Gym benchmarks) do not identify policy parameterization as a significant factor, largely because they exclusively consider Gaussian policies. In contrast, our work demonstrates that simply replacing the standard Gaussian actor with our discretized, regularized actor yields consistent and substantial performance gains, achieving state-of-the-art results across a wide range of continuous control tasks.
>
> **(3) "Regularized networks" terminology**
>
> We use the term regularized networks to refer to architectural choices that improve optimization stability, including residual connections and pre-layer normalization. These components act as implicit regularizers by stabilizing gradients and improving gradient signal quality.
>
> **(4) CMA-ES**
>
> CMA-ES means Covariance Matrix Adaptation Evolution Strategy. We will add the full name and reference [1] at first occurrence.
>
> **(5) Gradient variance curves**
>
> The upward trend reflects the later stage of training, where policies become more deterministic and exploration decreases. We extend training to 4× longer horizons on both environments, and the results are shown in [Figure 2](https://anonymous.4open.science/r/Rebuttal-figure-table-ICML26-BA46/rebuttal_only_figure.md).
> As the policy becomes more deterministic (i.e., lower entropy), the softmax operates in a higher-curvature regime, where small changes in logits induce larger changes in probabilities, increasing the variability of log-probability gradients. Importantly, despite this late-stage increase, the gradient variance of discrete policies remains several orders of magnitude lower than standard PPO throughout training.
>
> **(6) ManiSkill-RGB vs. ManiSkill-State**
>
> Hyperparameters.
> We adopt hyperparameters from the official CleanRL and ManiSkill implementations and keep them fixed across all methods for fair comparison.
>
> ManiSkill-RGB vs. ManiSkill-State.
> The ease of ManiSkill-RGB for some variants (e.g., StackCube-v1) is due to differences in representation learning. The CNN encoder used in RGB settings provides stronger inductive bias and feature extraction, which can simplify the learning problem and reduce sensitivity to actor design. In contrast, state-based inputs rely entirely on the policy network, making optimization more sensitive to architectural choices and leading to larger performance gaps across variants.
>
>
> **(7) Limitations of discrete policies**
>
> We agree and will expand the limitations section. Discretized policies can be challenging in high-dimensional action spaces, as the number of logits scales with both action dimension and bin count, increasing computational cost. Moreover, finer discretization requires predicting more precise distributions, which can exceed policy capacity and degrade performance. We will discuss these limitations and potential mitigations in the revision.
>
> **(8) Signal-to-noise ratio (SNR)**
>
> SNR is computed as the squared norm of the mean policy gradient divided by its variance across mini-batches. Intuitively, it quantifies the strength of the true gradient signal relative to stochastic noise. A higher SNR implies that gradient updates are more consistently aligned with the true gradient direction.
>
> However, SNR is not the only factor governing performance. Architectural choices (e.g., residual connections) can improve optimization and representation capacity beyond what SNR alone captures, so the best-performing models do not necessarily achieve the highest SNR. We will better contextualize the implications in the revision.
>
> [1] Nikolaus Hansen, “The cma evolution strategy: A tutorial,” arXiv preprint arXiv:1604.00772, 2016.

---

> > ### Author Rebuttal · Reviewer_1E8Z · 2026-04-03
> >
> > I appreciate the additional experiments and clarifications from the authors. Most of my concerns are resolved. My concern about novelty and significance remains, as the effectiveness of combining discrete actors with better network architectures is not surprising. Nevertheless, this alone is not a reason to prevent the paper from contributing potentially useful insights to the community.
> >
> > I'd be happy to increase my rating if the authors revise the SOTA claim to be more precise -- at least for Gym locomotion tasks, the proposed method is clearly not SOTA. As Reviewer nbkF pointed out, the PPO with RN-D has not even reached TD3's performance, let alone that of more recent work.

---

> > > ### Author Response · Authors · 2026-04-04
> > >
> > > We thank the reviewer for the thoughtful follow-up.
> > >
> > > **On novelty and significance.**
> > > We appreciate the perspective. While our method does not introduce a new architectural component, our main contribution is a unifying insight: policy parameterization induces different optimization objectives (e.g., cross-entropy vs. squared-error-like), which in turn affects gradient variance and training stability.
> > > We support this with both analysis and controlled experiments that isolate the role of parameterization. Our goal is to highlight a simple but underexplored design dimension that can consistently improve on-policy learning.
> > >
> > > **On the SOTA claim and positioning.**
> > > We fully agree and will revise the paper to clearly scope our claims to the on-policy setting and avoid broad SOTA statements. We will update the abstract and main text to ensure precise and accurate positioning.
> > >
> > > We thank the reviewer again for the constructive feedback and for recognizing the potential value of our work.

---

### Decision · Program_Chairs · 2026-04-30

**Decision:**

Accept (regular)

**Comment:**

This paper introduces an architectural framework for on-policy reinforcement learning that replaces standard Gaussian MLP actors with discretized categorical distributions and residual network structures. The authors called this method RN-D, and argue that this shift fundamentally alters the optimization landscape by moving from a weighted mean-squared error objective to a cross-entropy loss, which in turn reduces gradient variance and enhances training stability. Reviewers agree that the paper provides a rigorous and well-motivated empirical study, demonstrating that the proposed design choices significantly boost the performance of on-policy algorithms like PPO across various continuous control benchmarks.

The statistical significance of the results and the thorough investigation into how policy parameterization and network depth interact, have been particularly convincing to most of the reviewers.

Initial concerns regarding the novelty of using discretization and residual networks were a focal point of the discussion. For example, Reviewer 1E8Z and Reviewer nbkF raised some issues about novelty, that were solved by the authors' rebuttal.